# Prostanoid Signaling in Cancers: Expression and Regulation Patterns of Enzymes and Receptors

**DOI:** 10.3390/biology11040590

**Published:** 2022-04-13

**Authors:** Pavel V. Ershov, Evgeniy O. Yablokov, Leonid A. Kaluzhskiy, Yuri V. Mezentsev, Alexis S. Ivanov

**Affiliations:** Institute of Biomedical Chemistry, 10 Building 8, Pogodinskaya Street, 119121 Moscow, Russia; evgeniy.yablokov@ibmc.msk.ru (E.O.Y.); leonid.kaluzhskiy@ibmc.msk.ru (L.A.K.); yuri.mezentsev@ibmc.msk.ru (Y.V.M.); alexei.ivanov@ibmc.msk.ru (A.S.I.)

**Keywords:** prostanoids, GPCR, tumors, cancers, gene expression, regulation, TCGA, CPTAC, overall survival, disease prognosis, predictive value

## Abstract

**Simple Summary:**

Neoplastic processes are accompanied by the reprogramming of cell metabolism and alteration of the balance between endogenous bioregulators with signaling functions. Prostanoid signaling is a part of the global arachidonic acid pathway and is associated with cancer progression. It includes prostanoids (prostacyclin, thromboxane, and prostaglandins E_2_, F_2α_, D_2_, H_2_), prostanoid metabolizing enzymes, and receptors. We focused on a comparative systematic analysis of expression patterns of target genes, encoding prostanoid enzymes and receptors. We also addressed the possible ways of their regulation at different levels in normal and tumor tissues (expression of genes and proteins, mutation and copy number landscape, promoter methylation status, prediction of tissue-specific master regulators, microRNAs). The results of the systematic analysis made it possible to suggest models of regulation of differentially expressed prostanoid enzymes and receptors. The associations between gene expression signatures and patients’ overall survival rates were established which can be valuable for translational biomedicine.

**Abstract:**

Cancer-associated disturbance of prostanoid signaling provides an aberrant accumulation of prostanoids. This signaling consists of 19 target genes, encoding metabolic enzymes and G-protein-coupled receptors, and prostanoids (prostacyclin, thromboxane, and prostaglandins E_2_, F_2α_, D_2_, H_2_). The study addresses the systems biology analysis of target genes in 24 solid tumors using a data mining pipeline. We analyzed differential expression patterns of genes and proteins, promoter methylation status as well as tissue-specific master regulators and microRNAs. Tumor types were clustered into several groups according to gene expression patterns. Target genes were characterized as low mutated in tumors, with the exception of melanoma. We found at least six ubiquitin ligases and eight protein kinases that post-translationally modified the most connected proteins PTGES3 and PTGIS. Models of regulation of *PTGIS* and *PTGIR* gene expression in lung and uterine cancers were suggested. For the first time, we found associations between the patient’s overall survival rates with nine multigene transcriptomics signatures in eight tumors. Expression patterns of each of the six target genes have predictive value with respect to cytostatic therapy response. One of the consequences of the study is an assumption of prostanoid-dependent (or independent) tumor phenotypes. Thus, pharmacologic targeting the prostanoid signaling could be a probable additional anticancer strategy.

## 1. Introduction

Prostanoids (prostaglandins, prostacyclin, and thromboxane) are paracrine regulatory factors involved in signal transduction [1]. They have a systemic effect on many physiological processes under normal [2,3,4,5] and pathological conditions [6,7]. Determination of prostanoids in body fluids has diagnostic significance [8,9,10,11,12]. Prostanoid signaling is a functional cluster in the global arachidonic acid metabolism pathway [13] and consists of the enzymatic module (TBXAS1, PTGIS, PTGDS, PTGES1, PTGES2, PTGES3 and PRXL2B proteins) and the receptor module (TBXA2R, PTGIR, PTGDR, PTGFR, PTGDR2, PTGER1, PTGER2, PTGER3, PTGER4) (Figure 1).

Prostanoids are highly reactive factors and exported from “donor” cells via membrane transporters [14] and then, bind to surface receptors on “acceptor” cells [15,16,17]. The biomedical aspects of prostanoids in neoplastic transformation were thoroughly described in reviews [5,18,19] and confirmed in more recent experimental works [20,21,22,23,24,25,26,27,28,29,30,31,32,33,34,35,36,37] (Appendix B, Table A1). It can be noted that, firstly, correlations were established between the prostanoid levels in tumor tissues or in the blood (urinary) and the tumor growth, malignancy, and metastasis [22,23,24,26,28,29,30,32,33,34]. Secondly, prostanoids exert both tumor-suppressive and pro-neoplastic effects [25,37]. Thirdly, a correlation was found between the prostanoid content and tumor staging, histology, and patients’ survival rates [27,28,31]. Fourthly, inhibitors of prostanoid enzymes slowed down tumor growth and metastasis [21,35].

Data from integrative resources, for example, DisGeNET [38] and plenty of literature reports [39,40,41,42,43,44,45,46,47,48,49,50,51,52,53,54,55,56,57,58,59,60,61,62,63,64,65,66,67,68,69,70,71,72,73,74,75,76,77,78,79,80,81,82,83,84,85,86,87,88,89,90,91,92,93,94,95] (Appendix A), clearly demonstrate the significance of genes, encoding prostanoid enzymes and receptors in cancers. It is to be noted that there are reports on correlations of gene expression and methylation patterns in different tumors with disease prognosis [45,46,47,48,52,58,64,65,66,69,70,74,75,79,80,81,85,86,89,90]. Tumors with over-expressed or under-expressed prostanoid enzymes and receptors [41,42,43,49,55,69,78,82,91] could determine the benefits of pharmacological correction of this metabolic cluster [44,77]. Indeed, exploration of tumor-specific gene expression and regulation patterns of prostanoid enzymes and receptors contributed to the discovery of new candidate biomarkers and drug targets [7,96,97,98], which are of great value for translational medicine. However, we did not find a bioinformatic pan-cancer and/or cancer-specific analysis of the expression and regulation patterns of the entire pool of prostanoid enzymes and receptors. Nowadays, despite the increased number of literature reports on the prostanoid signaling field, there is still a lack of a sophisticated understanding of the impact of transcriptomic, proteomic, and metabolomic factors on tumor promotion or suppression.

The aim of this work was to perform a comprehensive analysis of the gene expression and regulation patterns in prostanoid signaling in the most common human cancers. According to the expression patterns, prostanoid-dependent and independent tumors were conditionally selected. Models of tumor-specific regulation of gene expression including a repertoire of master regulators, microRNAs associated with cancers (oncomiRs), the methylation status of gene promoters, protein-protein interactions, and modifying proteins were predicted. The prognostic values of several transcriptomics signatures were also revealed.

## 2. Materials and Methods

### 2.1. Gene Expression Analysis

Differentially expressed genes (DEGs) were selected from the Cancer Genome Atlas (TCGA) [99] twenty-four tumor datasets. “TCGA tumor” and “TCGA normal and GTEx data” datasets were compared using web-based tool GEPIA2 [100] (accessed on 10 November 2021) at *p*-value < 0.01 and cut-off value log_2_FC = 1. Datasets of hematologic malignancies were not analyzed. Gene co-expression analysis was performed using GEPIA2. The UALCAN browser [101] (http://ualcan.path.uab.edu/, accessed on 13 November 2021) was used to retrieve the list of genes, which are co-expressed with target genes at Pearson correlation coefficient ≥ 0.75.

The prognostic value of gene expression patterns was explored using pan-cancer Kaplan-Meier plotter [102] (https://kmplot.com/analysis/, accessed on 9 December 2021) with the following settings: follow-up period—“60 months”; number of patients with available clinical data—“more than 100”, auto-select—“best cut-off”. Overall survival analysis was performed using the TCGA data. The predictive value of differentially expressed genes was explored using the ROC-plotter server (http://www.rocplot.org/, accessed on 5 December 2021) with breast, ovarian, glioblastoma, and colorectal tumor datasets [103,104,105] with outlier exclusion in plot settings.

### 2.2. Mutation Status Analysis

The somatic mutation frequency of genes in different tumors was analyzed using eight pan-cancer cBioPortal cohorts with a total of 47,942 clinical cases [106,107].

The web-based tool muTARGET [108] (accessed on 31 October 2021) was used to predict associations between the mutation status of target genes in tumors and the expression of other genes with the following options: mutation type—“all somatic mutations”; *p*-value cut-off “0.05”; fold change cut-off “2”; FDR cut-off “no FDR filter”.

### 2.3. Over-Representation Analysis

The WEB-based GEne SeT AnaLysis Toolkit (Webgestalt server) [109,110] was used to enrich gene sets with KEGG (Kyoto Encyclopedia of Genes and Genomes), Reactome, and Gene Ontology (GO) terms with the following settings: multiple test adjustment—“Benjamini-Yekutieli FDR-controlling method (FDR < 0.1)”, minimal number of gene for category—“5”, redundancy reduction—“affinity propagation”.

### 2.4. Master Regulators

The prediction of tissue-specific master regulators (MRs) for each DEG was performed in the hTFtarget database [111] (http://bioinfo.life.hust.edu.cn/hTFtarget, accessed on 3 December 2021). hTFtarget accumulates pairs of transcription factors/genes from human ChIP-Seq data (7190 experiment samples of 659 transcriptional factors) in 569 conditions (399 types of cell line, 129 classes of tissues or cells, and 141 kinds of treatments). Tumor-specific gene expression patterns and cancer hallmarks of MRs were explored using GEPIA2 and Cancer Gene Census portal [112], respectively.

### 2.5. Tumor-Specific Non-Coding microRNAs (oncomiRs)

The Condition-Specific miRNA Targets database CSmiRTar [113] (http://cosbi4.ee.ncku.edu.tw/CSmiRTar/, accessed on 17 November 2021) was used to predict microRNA that is associated with cancer (oncomiRs) that can potentially participate in post-transcriptional regulation of target genes in prostanoid signaling. This server is linked to the five databases: DIANA-microT, miRanda.org, miRDB, Targetscan, and miRTarBase. The selection of predicted oncomiRs was performed by following settings: species—“human”, class of disease—“cancer”, average normalized score > 0.4 (for data availability at least in 3 of 5 supported databases). The OncoMir cancer database (OMCD) (www.oncomir.umn.edu/omcd, accessed on 21 November 2021) [114] was further used to study differential expression of oncomiRs with *p*-value threshold < 0.05 and fold change cut-off values > 2 or <0.5.

### 2.6. Promoter Methylation Status

The promoter methylation status of target genes was analyzed using the UALCAN browser. The beta-value indicates the level of DNA methylation ranging from 0 (unmethylated) to 1 (fully methylated). Different beta value cut-offs have been considered to indicate hypermethylation (beta value: 0.7–0.5) or hypomethylation (beta-value: 0.3–0.25) [115,116].

### 2.7. Protein Expression Analysis

Tumor-specific protein expression was explored using The Human Protein Atlas (HPA) portal version 21.0 [117] (https://www.proteinatlas.org/, accessed on 26 November 2021). The UALCAN browser was used to search for differentially expressed proteins (DEPs) within CPTAC datasets (breast, ovarian, colon, lung tumors, and uterine corpus endometrial carcinoma) [118]. Z-values represent standard deviations from the median across samples for the given cancer type. Log_2_ spectral count ratio values from CPTAC were first normalized within each sample profile, then normalized across samples.

### 2.8. Post-Translational Modifications

Modifying proteins carrying out post-translational modifications (PTMs) of prostanoid enzymes and receptors were retrieved from the Biogrid portal [119] (accessed on 10 January 2022). Only physical PPIs and matched subcellular localization profiles (according to the COMPARTMENTS database [120]) were used for the final selection of modifying proteins.

### 2.9. Protein-Protein Interaction Network Analysis

A list of known protein partners of PTGIS и PTGES3 was retrieved from STRINGdb v. 11.5 [121] (https://string-db.org/, accessed on 8 January 2022) with the following settings: organism—Homo sapiens; interaction sources—experiments; cut-off interaction score (combined score) = 0.4. The compartment database [120] (https://compartments.jensenlab.org/Search, accessed on 10 January 2022) was used for the subcellular localization analysis of proteins. Cytoscape software v. 3.8.2 was used for PPI network visualization [122].

### 2.10. Cluster Analysis of Gene Expression Matrix

Principal Component Analysis (PCA) and Classification and Regression Trees (CRT) methods were used to cluster DEGs in different tumors selected by |log_2_FC| tumor/normal > 1 and *p* < 0.01. For the PCA method, the gene expression matrix was prepared by assigning integer values (+2) or (−2) to the up- or down-regulated genes, respectively. Genes with |log_2_FC| < 1 were assigned zero value. The ClustVis web-based tool was used for PCA analysis and result visualization [123]. Data preprocessing options were the following: data transformation—“no transformation”; row scaling—“unit variance scaling”; PCA methods—“SVD with imputation”. Heat map options: clustering distance for rows—“Manhattan”; clustering methods for rows—“average”.

The CRT clusterization was performed by IBM SPSS Statistics v. 23 software with the Decision Tree algorithm. CRT classifies cases into groups of dependent variables based on the values of independent variables. CRT splits the data into segments that are as homogeneous as possible with respect to the dependent variable. A terminal node in which all cases have the same value for the dependent variable is a homogenous, “pure” node. The variables used were: Expression, dependent, and Gene, independent. Expression values were: NEGATIVE, POSITIVE, and NODIFF. The definitions of Expression values were as below. NEGATIVE: the gene expression level in the tumor datasets was decreased compared to the normal tissue datasets. POSITIVE: the gene expression level in the tumor datasets was increased compared to the normal tissue datasets. NODIFF: no changes were shown for the gene expression level in tumor and normal tissue datasets. Gene values were gene names. The Decision Tree procedure was used with the settings below: validation method—“cross-validation”, minimum cases in parent node—“100”, minimum cases in child node—“50”, level growth limits—“5”, impurity measure—“twoing”.

### 2.11. Other

The alignment of gene lists was performed using Venn diagrams (https://bioinformatics.psb.ugent.be/cgi-bin/liste/Venn/, accessed on 10 January 2022).

### 2.12. Data Mining

Molecular profiling data on 19 target genes, encoding prostanoid enzymes and receptors, from 24 solid tumors were analyzed using a data mining pipeline (Figure 2). The Cancer Genome Atlas (TCGA) portal was the main data-source for the gene expression and co-expression patterns as well as promoter methylation status. Tumor-specific protein expression data were retrieved from Clinical Proteomic Tumor Analysis Consortium (CPTAC) portal. Several web-based tools (GEPIA2, UALCAN, cBioPortal) were adapted for comparative statistical analysis of tumor and normal TCGA or CPTAC datasets to identify DEGs and DEPs. hTFtarget and CSmirTar portals were used for the prediction of master regulators and microRNAs for a subset of target genes, respectively. Finally, predictive and prognostic values of gene expression patterns were investigated using ROC-plotter and KM-plotter, respectively. Over-representation analysis was performed in the WebGestalt server.

## 3. Results

### 3.1. Gene Expression Analysis

The gene expression landscape of the prostanoid signaling in different tumors is shown in Figure 3. At least four groups of tumors can be conditionally distinguished according to a similar pattern of DEGs. In Figure 3, it was shown that group I consists of predominantly up-regulated DEGs in GBM, LIHC, PAAD, and THYM tumors. Group II has predominantly down-regulated DEGs in ACC, BRCA, KICH, KIRC, LUAD, PRAD, SKCM, THCA, UCES, and UCS tumors. Group III consists of both up- and down-regulated genes in COAD, ESCA, KIRP, LUSC, OV, READ, STAD, and TGCT. Group IV with HNSC and LGG tumors (not shown in Figure 3) does not contain DEGs.

Groups I and II with contrasted gene expression patterns are the most interesting for analysis. Up-regulation of genes, encoding enzymes, which produce pro-neoplastic prostanoids, can lead to tumor promotion and prostanoid-dependent tumor phenotype. Down-regulation of such genes may indicate tissue-specific responses to neoplastic transformation or cancer-driven metabolism reprogramming aimed to reduce the accumulation of tumor-suppressive prostanoids. To test these assumptions indirectly, correlations between the DEGs and disease prognosis were explored. We also considered the theory that the overexpressed genes (Figure 3) could be critical for tumor cell viability. In other words, how sensitive tumor cells are to knockouts and knockdowns of genes in prostanoid signaling. We analyzed the data of pan-cancer Clustered Regularly Interspaced Short Palindromic Repeats (CRISPR) and RNA interference (RNAi) screens on the DepMap portal [124]. A general rule is that the lower the gene effect score, the more its dependency in a cell line, so a score close to −1 corresponds to genes that can be essential for a cell line viability [124]. Appendix A shows the gene effect scores. Knockouts/knockdowns of several target genes with scores below −0.5 (an accepted cutoff value) can be related to tumor cell viability (Appendix A): *TBXAS1* (gallbladder adenocarcinoma, OCUG1 cell culture), *PTGDR* (lung cancer, CORL279), *PTGER4* (lymphoma, C8166) and *PTGES3* (brain cancer, ONS76). Incidentally, *PTGES3*, being the most common DEG in various tumors (Figure 3), tends to be an essential gene in tumor cell lines (Appendix A). At the same time, there is an absence of a significant effect in cell lines caused by the “separate switching off” of most of the target genes. Thus, it allows us to consider these genes as non-essential for tumor cell viability.

Cluster analysis was used to find outmatched expression patterns of target DEGs within a set of tumors. Comparing the results obtained by two clustering methods, the Classification and Regression Trees (CRT) (Appendix C, Figure A1) and the Principal Component Analysis (PCA) (Figure 4), subclusters with matched gene expression patterns (*PTGIS-PTGDS*, *PTGES2-PTGDR2*, *TBXA2R-PTGER4-PTGER1-PTGER-PTGDR*, and *PTGES3-PRXL2B*) were identified.

PCA method indicated three large clusters. The first and second expression clusters consist of genes encoding prostanoid enzymes: *TBXAS1*, *PTGIS*, *PTGDS*, *PTGES3*, *PRXL2B*, and *PTGES*, *PTGES2*, *AKR1C3*, *CBR1*, *CBR3*, respectively. The third cluster is represented exclusively by genes encoding prostanoid receptors. As for PCA clusters, high Pearson correlation coefficients of gene expression within the cluster were only in case of *AKR1C3*, *CBR1*, *CBR3* genes (r = 0.64 − 0.77, *p*-value < 0.05) in LUSC tumor as well as CBR1 and CBR3 (r = 0.72 − 0.86, *p*-value < 0.05) in ESCA tumor. There was no significant correlation in respective normal tissues. In the third PCA cluster, co-expression between the *PTGIR* and *TBXAR2* genes (r = 0.67 − 0.82, *p*-value < 0.01) was found in ESCA, KIRP, LICH and READ tumors. Other pairs of co-expressed genes encoding prostanoid receptors were as follows: *PTGIR-PTGER2*, r = 0.72 − 0.75, TGCT tumor; *PTGER4-PTGER2*, 0.68–0.78, SKCM tumor; *PTGDR-PTGER2*, r = 0.66 − 0.74, SKCM tumor; *PTGDR-PTGER3*, 0.59–0.67, PAAD tumor.

Next, we searched for other genes whose expression patterns correlate with the target genes in each of the three PCA clusters (Figure 4). It should be said that the total number of found co-expressed genes for the first cluster was about 10 times higher than for the second cluster. Gene sets co-expressed with *TBXAS1* and *PTGDS* in a subset of tumors (KIRC, LIHC, GBM, PCPG, OV, ACC, LGG, UVM, ESCA, CHOL) are enriched in immune reaction and phagocytosis pathway terms (Appendix A). Gene sets co-expressed with *PTGIS* in DLCA, CHOL, COAD, READ, PRAD, TGCT tumors, and *PTGES* in ACC tumors are enriched in “muscle contraction” and “extracellular matrix organization” terms. Gene sets co-expressed with *PTGES3* in THCA and SKCM tumors are enriched with the “mRNA processing” term (Appendix A). Gene sets co-expressed with genes, encoding prostanoids receptors TBXA2R, PTGFR, and PTGER3, are involved in immune responses and structural processes (elastic fibers formation, collagen polymerization/depolymerization) as well as the organization of the extracellular matrix (Appendix A) in COAD and TGCT tumors. These findings indicate the leading cellular processes that accompany prostanoid signaling in different tumors. These terms are in good agreement with the known effects of prostanoids on blood vessel smooth muscle proliferation in the regulation of cardiovascular homeostasis [125,126], actin cytoskeleton reorganization via stimulation of stress fiber formation [127], and the immune response modulation [127,128,129]. Enrichment analysis also shows the association of prostanoid enzymes and receptors with focal adhesion kinase 1 (PTK2), which participates in the neoplastic transformation via activation of Wnt/β-catenin signaling [130].

### 3.2. Regulation Patterns of Differentially Expressed Genes

#### 3.2.1. Promoter Methylation

Differences in the transcript accumulation in normal and tumor tissues may be the result of several competing factors at the transcriptional and post-transcriptional levels such as the methylation status of gene promoters, combination of transcriptional master regulators, and transcript stability. Analysis of promoter methylation status of the target genes show *PTGDR*, *PTGER3*, *PTGIR*, and *TBXA2R* down-regulation in a number of tumors which may be partly due to the elevation of promoter methylation (Appendix A). On the other hand, for up-regulated *CBR3* and *AKR1C3* genes in LUSC tumors, there is an agreement with promoter hypomethylation. The down-regulation of *PTGDS* in ESCA and HNSC tumors was accompanied by a statistically significant decrease in promoter methylation while remaining in the hypermethylation range.

#### 3.2.2. Master Regulators

Transcriptional master regulators (MRs) mean DNA-binding and chromatin remodeling proteins [131], which are capable of regulating the expression of genes-of-interest. We selected only tissue-specific MRs that were predicted for ≥80% of all the target DEGs. Table 1 shows the distribution of 21 MRs that could potentially be responsible for the differential expression of target genes in tumors. Master regulators such as BRD4, CTCF, EP300, FOXA1, and SPI1 are expressed in breast, brain, colorectal, kidney, pancreatic, prostate, skin, and stomach tumors (Table 1) enriched with cancer hallmarks and encoded by cancer driver genes. It is noteworthy that most of the found MRs are overexpressed in tumors. This speaks in favor of the involvement of the predicted MRs in the neoplastic transformation that is also stressed by the participation of AR, EP300, MAX, RELA, SP1, and SPI1 proteins in cancer-related pathways (Appendix A).

#### 3.2.3. OncomiRs

A list of regulatory oncomiRs was also predicted for all of the target DEGs (Appendix A) and oncomiRs expression patterns in tumors were explored. The number of oncomiRs for prostanoid receptors was much higher than for prostanoid enzymes, decreasing in the following line: *PTGER4-PTGER3-PTGER2-PTGDR-TBXAR2-PTGFR*. Appendix A shows the tumor/normal ratios of “master oncomiRs”, which can potentially regulate at least two target transcripts. It turned out unexpectedly that some oncomiRs were predicted as universal post-transcriptional regulators for both prostanoid enzymes and receptors such as miR-149-3p (*PRXL2B*, *PTGES*, *TBXA2R*), miR-19a-3p (*PTGES3*, *PTGER2*), miR-338-5p (*PTGES3*, *PTGDR*), miR-421 (*PTGES3*, *PTGER2*); miR-423-5p (*PRXL2B*, *PTGFR*), miR-508-5p (*PTGES*, *TBXA2R*). Most oncomiRs are up-regulated in tumors as compared to corresponding normal tissues. Findings on changes in tumor-specific expression of oncomiRs and target transcripts are shown in Figure 5.

Thus, it can be assumed that the accumulation of oncomiRs may be in inverse proportion with that of target transcripts. This can be traced in contrasting pairs such as miR-19a/*PTGER2* (pancreatic tumor); miR-421/*PTGER2* (uterine tumor); miR-590/*PTGFR* (uterine and breast tumor); miR-20a/*PTGER3* (uterine tumor) (Figure 5).

### 3.3. Protein Expression Patterns

According to The Human Proteome Atlas (HPA), positive immunohistochemical staining (IHC) is shown for the majority of prostanoid enzymes in tumors (Appendix A). IHC protein expression data on prostanoid receptors, with the exception of PTGER4, are not yet available in HPA (Appendix A). This group can be characterized as low expressed in normal and tumor tissues. *PTGDR*, *PTGDR2*, *PTGER1*, *PTGER3*, and *PTGFR* genes have the median expression value of about 1 FPKM (Fragments Per Kilobase Million), while other genes are expressed in the range of 1–5 FPKM, regardless of cancer specificity. A concordance between the transcripts and protein accumulation in normal and tumor tissues was performed by comparing the TCGA and CPTAC data in BRCA, COAD, LUAD, OV, and UCES tumors (Table 2). From Table 2, it follows that there is a concordance between the transcripts and total protein accumulation. However, up-regulation of *PRXL2B* is not accompanied by an increase in the protein content in BRCA and OV tumors. There is an inverse relationship between the up-regulated *PRXL2B* and *PTGDS* genes and the protein accumulation in COAD and OV tumors, respectively. The general observation is that a decrease in mRNA accumulation leads to a decrease in total protein, which is quite logical. It is assumed that when transcript level increases, but protein content remains unchanged or reduced, there is a translational and/or post-translational regulation.

### 3.4. Prognostic Value of Transcriptomic Signatures

We explored the target genes, encoding the prostanoid enzymes and receptors, in terms of their disease prognostic value. Survival analysis was performed, and all relevant results obtained are presented in Table 3. In particular, Appendix A shows overall survival curves in groups with high and low gene expression (without subgroup analysis) followed by an assessment of tumor-specific signatures (Table 3). A “pure” tumor specificity of signatures is achieved in the subgroups stratified by gender, stage, grade, and mutation burden. This, however, is balanced by a reduction of clinical cases in a subgroup and the power of a statistical test. Most of the transcriptomic signatures (without subgroup analysis) still show acceptable tumor specificity with the exception of the signature containing PTGDS, CBR3, PTGIR, PTGFR, PTGDR2, and PTGER3 genes in HNSC tumor, which is similar to other tumors (BRCA, CESC, LUAD, SARC, and UCES).

### 3.5. Predictive Value of Prostanoid Enzymes and Receptors Genes

We explored the associations between gene expression of each target gene and responses to anticancer drug treatment of breast, ovarian, glioblastoma, and colorectal tumors using the Receiver Operating Characteristic plotter (ROC-plotter) [103]. It was found that only for breast or ovarian tumors, ROC curves for (*PTGIS*, *PTGES*, and *PTGER4*) or (*TBXAS1*, *PTGES*, *TBXA2R*, and *PTGDR2*), respectively, had Area Under Curve (AUC) values > 0.75 at the tumor/normal fold changes ≥ 2 (Appendix A). *PTGIS*, *PTGES*, and *PTGER4* up-regulation in the group of responders with HER2-positive breast cancer, known by its aggressive behavior [132,133], is associated with complete tumor response to treatment with fluorouracil, epirubicin, and cyclophosphamide (FEC). On the other hand, the same amplitude of down-regulation of *PTGES* and *PTGDR2* gene expression in the group with serous ovarian cancer grade G3 correlates with relapse-free survival at 6 months and the response to taxanes and platinum treatment, respectively (Appendix A). Thus, we observed the associations between the gene expression patterns and responses to cytostatic therapy, while there were no associations in the case of targeted therapy (Trastuzumab, Tamoxifen, Avastin, and aromatase inhibitors).

### 3.6. Mutation Status Analysis

It was found that the highest frequency of somatic mutations in target genes was in melanoma (cut-off = 0.5%, *n* > 100). Next, cBioPortal cohorts “Skin Cutaneous Melanoma TCGA PanCancer Atlas” (*n* = 448), “TCGA Firehose Legacy” (*n* = 479) and “DFCI Nature Medicine 2019 metastatic melanoma” (*n* = 144) were analyzed. Mutation frequency rates at 5–6%, 5–7%, 2.8–5%, 3–4%, 7–9%, 2.1–5% were observed in *TBXAS1*, *PTGIS*, *AKR1C3*, *PTGDR*, *PTGFR* and *PTGER3* genes, respectively. muTARGET web-based tool allowed us to predict genes, which expression patterns can be associated with genetic polymorphism in group #1 (*TBXAS1*, *PTGIS*, and *AKR1C3*) or group #2 (*PTGDR*, *PTGFR*, and *PTGER3*) in melanoma. Thirty six down- and three up-regulated genes were predicted for group #1, and 14 and 6 for group #2, respectively (Appendix A). Down-regulated genes were enriched with the following GO terms: “extracellular region”, “extracellular exosome”, “heparin binding” (*BMP7*, *FMOD*, *PCSK6*, and *THBS4*), “cell adhesion” (*ATP1B2*, *BCAN*, *CNTN4*, *PKP1*, *RELN*, *ROBO2*, *SPON1*, *SVEP1*, and *THBS4*), “epidermal growth factor-like domain” (*BCAN*, *PCSK6*, *RELN*, *SCUBE3*, *SVEP1*, and *THBS4*), “fibronectin type III domain” (*CNTN4*, *ROBO2*, *SDK2*, and *SORL1*). A subset of up-regulated genes was represented by *ILDR2*, *CDH2*, *RNF128*, *GBP1*, *PDCD1LG2*, *ALDH1A2*, *ADAMTS14*, *RN7SK*, and *RGS5*, however, statistically significant GO terms were not found.

## 4. Discussion

Pan-cancer analysis showed several interesting gene expression patterns in prostanoid signaling that were used to model tissue-specific regulation patterns (Appendix A). Transcriptomic data were retrieved from the TCGA portal, which actually contains data on most tumor types and subtypes. At the same time, the availability of proteomic information in the CPTAC portal is more limited. For this reason, we compared the gene expression patterns of target genes, their predicted master regulators, and oncomiRs in several tumors. It could be suggested that the up-regulation of *AKR1C3*, *CBR1*, and *CBR3* genes from PCA cluster 2 (Figure 4) in LUSC tumors may occur due to promoter hypomethylation and gene expression changes of master regulators *FOXA2*, *LMNB1*, *SPI1*, miR-511 (Appendix A). In contrast, down-regulation of co-expressed genes *CBR1* and *CBR3* in ESCA tumors can be associated with elevated accumulation of miR-335 (for *CBR1*) and a decrease in promoter methylation status similar to that in LUSC. The up-regulation of all ten genes, encoding prostanoid enzymes, in PAAD tumors compared to normal tissue is noteworthy. This expression pattern is not found in any other tumors, except for THYM tumors, where only seven of ten genes are up-regulated. It follows from Appendix A that at least half of DEGs are characterized by an increase in copy number variations (amplification type) indexed in the cBioPortal UTSW Nat. Commun. 2015 cohort. We have found no statistically significant changes in the promoter methylation status in PAAD tumors compared to normal tissue. However, up-regulated *PTGDS*, *AKR1C3*, and *CBR3* genes were hypermethylated, an epigenetic situation that also occurs in tumors [134]. The accumulation of *PTGES3* and *PRXL2B* transcripts in tumor tissue correlates with a simultaneous reduction of oncomiRs miR-223, miR-19a, miR-605, and miR-486, miR-211, miR-423, respectively. It should be noted that some of those are tumor suppressors [135,136,137]. Up-regulation of ten DEGs in PAAD tumor is in concordance with the up-regulation of master regulators *CTCF*, *IRF1*, and *KLF4*, while *POLR2A* and *STAG1* remain unchanged. It is well known that transcriptional activation of master regulators is critical for tumor progression, in particular, for pancreatic [138] and colorectal cancers [139,140].

Prostacyclin, a metabolite produced by prostacyclin synthase (PTGIS), is historically believed to exert tumor-suppressive effects [141,142] and lowers its level along with down-regulation of PTGIS associated with an aggressive tumor phenotype and a poor disease prognosis [43]. Figure 3 shows a simultaneous decrease in the expression of both *PTGIS* and *PTGIR* in eight tumor types, and therefore, we analyzed the possible causes of such changes (Appendix A). It can be pointed out that down-regulation of *PTGIS* in KIRP, LUAD, THCA, and UCES tumors is accompanied by an increase in miR-34a levels, which mainly plays a considerable role in inhibiting tumor progression in thyroid tumors [143]. In LUSC tumors, there is a decrease in miR-34c, which, like miR-34a, possesses antitumor activity [144]. On the other hand, miR-326 expression is suppressed in lung cancer tissues. This oncomiR, as shown in [145], inhibits lung cancer cell proliferation, and colony formation and provokes apoptosis. It should also mention that the down-regulation of *PTGIS* is comparable to the lowering of a corresponding protein product in LUAD and UCEC tumors (Table 2). As for the predicted transcriptional master regulators of PTGIS and PTGIR (Appendix A), there are fundamentally distinct tumor-specific patterns. Transcript and total protein accumulation of master regulator ZBTB7A as well as PTGIS and PTGIR were reduced in UCES tumors, which was markedly related to the stage and prognosis of this tumor type [146]. Thus, the down-regulation of PTGIS and PTGIR in eight different tumors in our model may be due to the contribution of master regulators and oncomiR combinations at the transcriptional and post-transcriptional levels under conditions of unchanged promoter methylation status and copy number variations (deletion type) of target genes.

### Protein-Protein Interactions and Post-Translational Modifications

Since some of the prostanoids are quite metastable (short-lived) metabolites, spatial clustering or compartmentalization of prostanoid enzymes can be required. It is realized through either direct PPIs between enzymes or the involvement of common protein partners as well as post-translational modifications (PTMs). Previously, using the affinity purification and mass-spectrometry approach, we revealed that the PTGES3 protein could be a potential protein partner of PTGIS [147]. PPIs subnetworks with PTGIS and PTGES3 and their protein partners retrieved from STRINGdb are shown in Appendix A. Overrepresentation analysis (ORA) indicates that a subset of the PTGIS’s protein partners is enriched with steroid and cholesterol biosynthesis (FDFT1, HSD17B7, LSS, SC5D proteins) pathway terms. Functional “bridges” between cholesterol synthesis and prostanoid pathways (changes in prostacyclin levels in the presence of statins), as described in [148,149], can be mediated via modulation of *PTGIS* gene expression. But so far, we have not found studies that would evidence the functional value of PPIs with enzymes involved in prostanoid and cholesterol synthesis. The PTGES3′s protein partner subset is enriched with pathway terms such as “protein processing in endoplasmic reticulum”, “pathways in cancer” (KEGG); “cellular responses to external stimuli”, “aryl hydrocarbon receptor signaling” and “TNF alpha Signaling Pathway” (Reactome). In addition, comparing the data obtained in [147] and STRINGdb, we found that at least heat shock 70 kDa protein 4L (HSPA4L, Uniprot ID: O95757) and calreticulin (CALR, Uniprot ID: P27797) with chaperone activity are common protein partners of PTGIS and PTGES3 proteins.

Amino acid sequences of prostanoid enzymes and receptors contain multiple sites for reversible post-translational modifications such as ubiquitination, phosphorylation/dephosphorylation, and glycosylation. In that context, the gene expression patterns of modifying proteins (ubiquitin ligases, protein kinases/phosphatases, and glycosyltransferases) were analyzed. The spectrum of potential modifying proteins, which physically interact with prostanoid enzymes and receptors, and their tumor-specific gene expression patterns are shown in Appendix A. Up-regulation of ubiquitin-protein ligases SIAH2, MARCH2, MARCH3, UBE2W, OTUB1, and VHL, which regulate the stability of mature proteins, as well as the protein kinases STK24, MAP4K1, MAP4K4, PRKCD, CSK, PINK1, PRKAB1, and STK39, was found. It is known that mitogen-activated protein kinases (MAP4K1 and MAP4K4) and the insulin resistance pathway (PRKCD and PRKAB1) may be associated with tumor progression through modulation of gene expression responsible for cell cycle, proliferation, and growth [150]. We have schematically shown in Figure 6 the associations between gene expression patterns of CBR1 and PTGIR proteins and their modifying enzymes.

These examples demonstrate tumor-specific multiple modes of post-translational regulation for prostanoid enzymes and receptors, so the presence of “active combinations” of modifying enzymes depending on their expression levels in tumors can be suggested. However, the direct involvement of modifying proteins in post-translational modifications of target enzymes or receptors is still not sufficiently studied. It is known that PTGES protein stability is positively regulated through interaction with ubiquitin-specific peptidase 9 X-Linked (USP9X) [57]. Post-translational regulation via phosphorylation has been described for PTGES [151]. Prostanoid receptors have rather long cytoplasmic tails with potential phosphorylation sites [152], and protein kinase C-dependent phosphorylation has been described for the prostacyclin receptor [153].

The limitations of the study are related to the use of publicly available data from TCGA, CPTAC, and other repositories and web-based tools for the analysis of datasets. The results of the study have more fundamental rather than translational value. The identified transcriptional signatures, with the participation of prostanoid signaling genes with differential expression in tumor/normal tissues, are exploratory. Thus, to further establish the clinical relevance of such signatures, additional rounds of experimental validation should be required.

## 5. Conclusions

We investigated the highly heterogeneous gene and protein expression landscape of prostanoid enzymes and receptors in 24 different tumors and suggested the models of tumor-specific regulation. Nine bioinformatic web-based tools (GEPIA2, UALCAN, cBioPortal, hTFtarget, CSmirTar, ONCOmir, muTARGET, Biogrid, and ClustVis) were used for the analysis of differentially expressed genes, proteins, microRNAs, methylation and mutation patterns, as well as protein-protein interactions. Four groups of tumors were prioritized according to the profiling of the entire pool of differentially expressed target genes. The high correlation of co-expression was shown in the sub-cluster with *AKR1C3*, *CBR1*, and *CBR3* genes. Down-regulation of *PTGDR*, *PTGER3*, *PTGIR*, and *TBXA2R* genes in a number of tumors can be linked with promoter methylation status. Tissue-specific master regulators BRD4, CTCF, EP300, FOXA1, and SPI1, overexpressed in tumors, were found for target genes. Predicted microRNAs such as miR-149-3p, miR-19a-3p, miR-338-5p, miR-421, miR-423-5p, and miR-508-5p can be involved in the post-transcriptional regulation of at least two different target RNAs. The highest concordance between expression data of TCGA and CPTAC databases was achieved for *PTGIS* and *PTGES* genes in four tumors: BRCA, COAD, LUAD, and UCES. Mutation frequency of *TBXAS1*, *PTGIS*, *AKR1C3*, *PTGDR*, *PTGFR* and *PTGER3* genes in melanoma were 5–6%, 5–7%, 2.8–5%, 3–4%, 7–9%, 2.1–5%, respectively. One of the conclusions of the study is the assumption of the presence of prostanoid-dependent tumor phenotypes. It can be demonstrated by the total up-regulation of prostanoid synthesis enzymes in GBM, PAAD, and THYM tumors. Down-regulation of the *PTGIS* and *PTGIR* genes in eight different tumors may be associated with a more aggressive tumor phenotype due to the abolishment of prostacyclin’s tumor-suppressive effects. Models of CBR1 and PTGIR post-translational regulation models were mediated via neddylation and ubiquitination/deubiquitination as well as phosphorylation depending on tumor types. We have found several multigene cancer-specific transcriptomic signatures (in BLCA, CESK, SARC, LUAD, LIHC, and KIRP tumors) associated with patients’ overall survival rates (prognostic value). There are associations between the expression pattern of five target genes and the tumor response to cytostatic therapy. For example, differential expression of the PTGES gene was predictive in BRCA and OV tumors. From our point of view, for the first time, a systemic pan-cancer analysis of the molecular features of the expression of genes involved in prostanoid signaling was carried out. The new results obtained will be contributed to the insight into prostanoid signaling in a cancer context.

## Figures and Tables

**Figure 1 biology-11-00590-f001:**
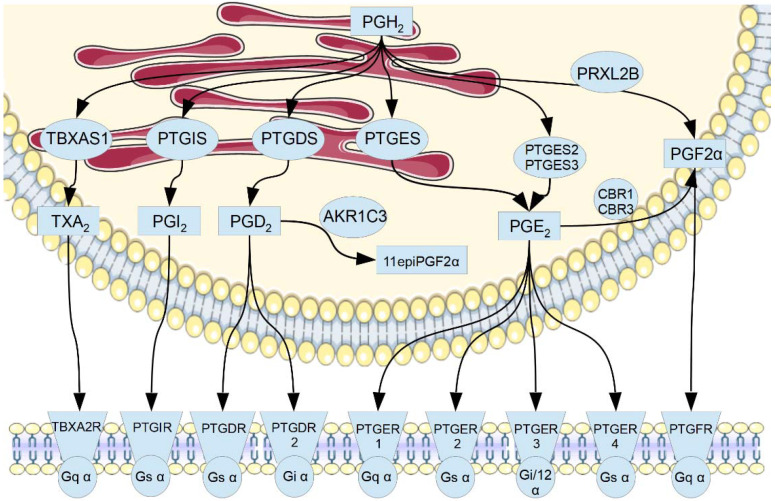
A scheme of prostanoid signaling: prostanoid-metabolizing enzymes (ellipses), G-protein-coupled prostanoid receptors (trapezes), type of G-subunit (circles), and prostanoids (rectangles). Proteins are shown according to their subcellular localization (endoplasmic reticulum, cytosol, and plasma membrane). Proteins’ and prostanoids’ names correspond to the list of abbreviations. Membrane and endoplasmic reticulum image templates were obtained from https://smart.servier.com/ (accessed on 3 February 2022).

**Figure 2 biology-11-00590-f002:**
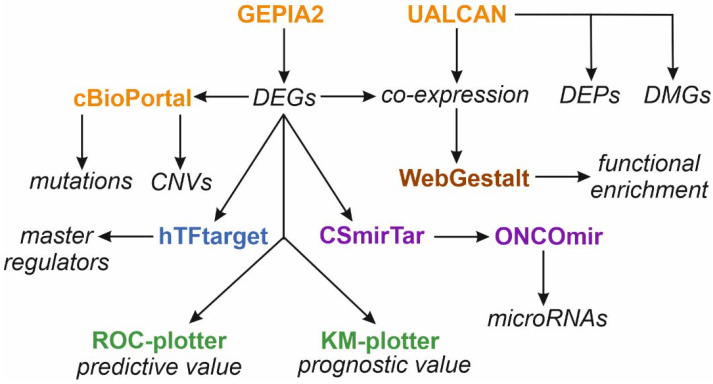
A flowchart of data mining using web-based bioinformatic tools: GEPIA2, UALCAN, and cBioPortal were used for the analysis of The Cancer Genome Atlas (TCGA) and Clinical Proteomic Tumor Analysis Consortium data (CPTAC); WebGestalt—WEB-based GEne SeT AnaLysis Toolkit; hTFtarget—database for regulations of human transcription factors and their targets; CSmirTar—Condition-Specific miRNA Targets database; ONCOmir—OncoMir Cancer Database. ROC-plotter—ROC-plotter server; KM-plotter—Kaplan-Meier plotter server. Abbreviations: DEGs—differentially expressed genes; DEPs—differentially expressed proteins; DMGs—differentially methylated genes; CNVs—copy number variations; mutations—cancer-specific mutation frequency of target genes.

**Figure 3 biology-11-00590-f003:**
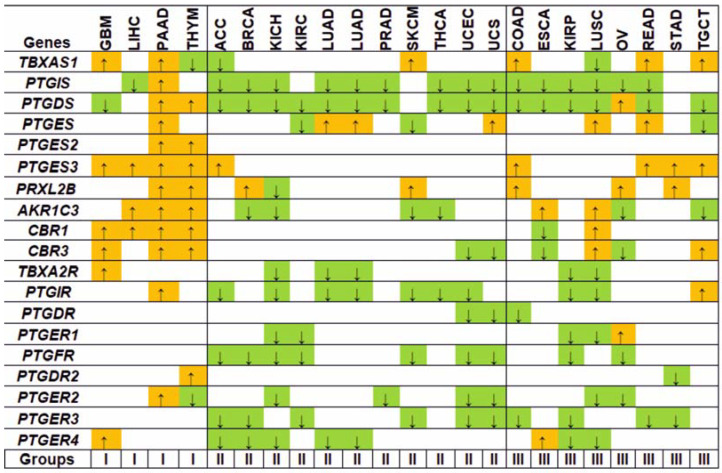
A landscape of differentially expressed genes, encoding prostanoid-metabolizing enzymes (*TBXAS1*, *PTGIS*, *PTGDS*, *PTGES*, *PTGES2*, *PTGES3*, *PRXL2B*, *AKR1C3*, *CBR1*, *CBR3*) and receptors (*TBXA2R*, *PTGIR*, *PTGDR*, *PTGFR*, *PTGDR2*, *PTGER1*, *PTGER2*, *PTGER3*, *PTGER4*), in different tumors. Statistically significant changes (fold change cutoff = 2) in tumor/normal tissues are shown by arrows. “Groups” correspond to groups of tumors distinguished according to a similar pattern of DEGs. Up- and down-regulated genes are highlighted with orange and green colors, respectively. Genes’ and tumors’ names correspond to the list of abbreviations.

**Figure 4 biology-11-00590-f004:**
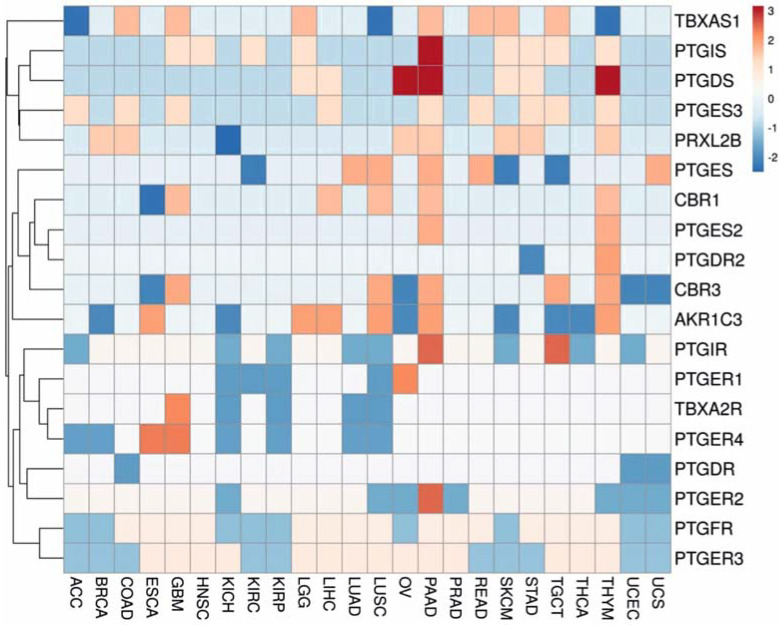
Principal component analysis of differentially expressed genes, encoding prostanoid metabolizing enzymes and prostanoid receptors, in different tumors; color scale shows cluster distances.

**Figure 5 biology-11-00590-f005:**
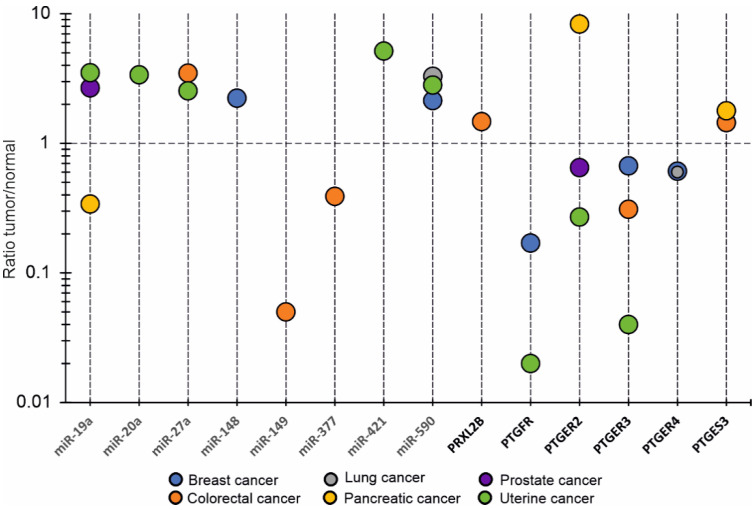
Comparative analysis of expression patterns of oncomiRs and genes, encoding prostanoid-metabolizing enzymes and prostanoid receptors, in different tumors. Genes’ names correspond to the list of abbreviations.

**Figure 6 biology-11-00590-f006:**
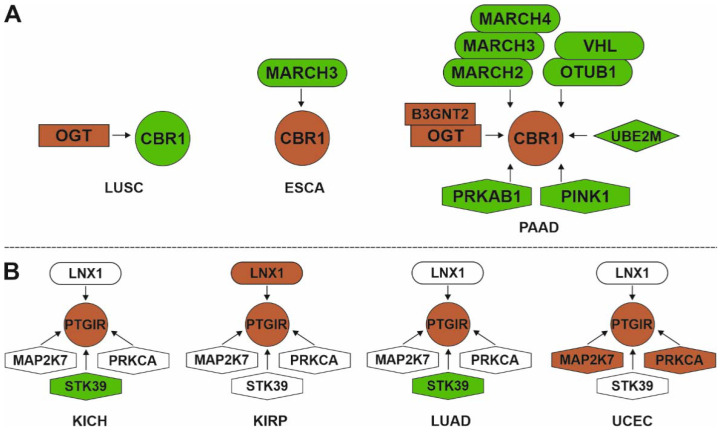
Associations between tumor-specific gene expression patterns of modifying enzymes and proteins in prostanoid signaling: (**A**)—CBR1; (**B**) PTGIR. Up- and down-regulated genes are highlighted with green and red colors, respectively. Ubiquitination enzymes, protein kinases, neddylation, and glycosylation enzymes are represented in oval, polygon, diamond, and rectangle shapes, respectively. Abbreviation: B3GNT2—(N-acetyllactosaminide beta-1,3-N-acetylglucosaminyltransferase 2); CBR1—(carbonyl reductase [NADPH] 1); LNX1—(E3 ubiquitin-protein ligase LNX); MAP2K7—(dual specificity mitogen-activated protein kinase 7); MARCH2—(E3 ubiquitin-protein ligase MARCHF2); MARCH3—(E3 ubiquitin-protein ligase MARCHF3); MARCH4—(E3 ubiquitin-protein ligase MARCHF4); OGT—(UDP-N-acetylglucosamine--peptide N-acetylglucosaminyltransferase 110 kDa subunit); OTUB1—(uUbiquitin thioesterase OTUB1); PINK1—(serine/threonine-protein kinase PINK1, mitochondrial); PRKAB1—(5′-AMP-activated protein kinase subunit beta-1); PTGIR—(prostacyclin receptor); STK39—(STE20/SPS1-related proline-alanine-rich protein kinase); UBE2M—(NEDD8-conjugating enzyme Ubc12); VHL—(von Hippel-Lindau disease tumor suppressor).

**Table 1 biology-11-00590-t001:** Tissue-specific master regulators of the expression of genes, encoding prostanoids enzymes and receptors in different cancers.

Cancer	Tissue-Specific Master Regulators
Breast cancer	***BRD4*** ▬, ***CTCF*** ▬, ***EP300*** ▬, ***FOXA1*** ▲▲, *SPI1* ▬
Brain cancer	*POLR2A*▬, *SPI1 *▲▲▲
Colorectal cancer	***CTCF*** ▬, *SP1*▬
Esophageal cancer	*KDM4C* ▬
Kidney cancer	*AR*▲, *RNF2*▬, *SPI1* ▲, *ZNF263*▬
Liver cancer	*FOXA2*▬, *HNF4A*▬, *MAX*▬
Lung cancer	*LMNB1*▲, *MAZ*▬, *RELA*▬, *SPI1* ▼
Pancreatic cancer	***CTCF*** ▲, *POLR2A*▬
Prostate cancer	*AR*▬, ***FOXA1*** ▲
Skin cancer	***CTCF*** ▬, *SPI1* ▲
Stomach cancer	*KLF5*▲▲▲, *SPI1* ▲
Uterine cancer	*NFIC*▼, *ZBTB7A* ▼

Note: Cancer driver genes and cancer hallmarks are highlighted with bold and underlined, respectively. Up- and down-regulated genes (1 < log2FC < 2, tumor/normal) are marked with arrows ▲ and ▼, respectively; 2 < log2FC < 3, ▲▲; log2FC > 3, ▲▲▲. No significant changes of gene expression (▬).

**Table 2 biology-11-00590-t002:** A concordance between transcript and protein accumulation in tumors.

Tumor	BRCA		COAD		LUAD		OV		UCES	
Gene Names	TCGA	CPTAC	TCGA	CPTAC	TCGA	CPTAC	TCGA	CPTAC	TCGA	CPTAC
*TBXAS1*	▬	▬	▲	▲	▬	▬	▬	▬	▬	▬
*PTGIS*	▼	▼	▼	▼	▼	▼	▼	▼	▼	▼
*PTGDS*	▼	▼	▼	▼	▼	▼	▲	▼	▼	▼
*PTGES*	▬	▬	▬	▬	▲	▲	▬	▬	▬	▬
*PTGES3*	▬	▬	▲	▲	▬	▬	▬	▬	▬	▬
*PRXL2B*	▲	▬	▲	▼	▬	▬	▲	▬	▬	▬
*AKR1C3*	▼	▼	▬	▬	▬	▬	▼	▼	▬	▬
*CBR3*	▬	▬	▬	▬	▬	▬	▼	▼	▼	▼

▲ and▼, significant (*p*-value < 0.05) increase and decrease in transcript (TCGA datasets) or protein (CPTAC datasets) levels (tumor/normal tissue), respectively; ▬ no significant changes.

**Table 3 biology-11-00590-t003:** Prognostic value of transcriptomics signatures of genes, encoding prostanoids enzymes and receptors.

Gene Expression Signature	Tumors, Subgroups	Hazard Ratio (CI), Logrank *p*-Value	Quartile, Survival (Months) Low-High Expression Cohorts	Signature Specificity Compared to Different Tumors
*PTGIS*, *PTGDS*, *PTGFR*, *PTGER3*	BLCA	2.1 (1.4–3.2), 9.2 × 10^−5^	Q1, 21–12	KIRP
	BLCA, male gender	2.5 (1.6–4.0), 7.3 × 10^−5^	Q1, 22–12	not found
	BLCA, stage 3	2.2 (1.0–4.7), 0.034	Q1, 21–13	not found
*PTGDS*, *CBR3*, *PTGIR* *PTGFR*, *PTGDR2*, *PTGER3*	HNSC	0.6 (0.4–0.8), 0.00032	Q1, 26–59	BRCA, CESC, LUAD, SARC, UCES
	HNSC, male gender, high mutation burden	0.4 (0.2–0.8), 0.0058	Q1, 13–47	not found
	HNSC, male gender, low mutation burden	0.5 (0.3–0.8), 0.0031	Q1, 12–23	not found
	HNSC, stage 3	0.2 (0.1–0.6), 0.00058	Q1, 11–57	not found
*PTGDS*, *AKR1C3*, *CBR1*, *CBR3*, *PTGDR*, *PTGDR2*, *PTGER2*, *PTGER4*	CESK	0.5 (0.3–0.8), 0.0067	Median, NA–NA	SARC
	CESK, female gender, white race	0.4 (0.2–0.8), 0.0027	Q1, 21–42	not found
*TBXAS1*, *PTGDS*, *AKR1C3*, *PTGIS*, *CBR3*, *TBXA2R*, *PTGDR*, *PTGFR*, *PTGER3*, *PTGER4*	SARC	0.5 (0.3–0.7), 00026	Q1, 16–37	BRCA, CESK,
	SARC, high mutation burden	0.4 (0.2–0.7), 0.00043	Q1, 11–37	not found
	SARC, low mutation burden	0.4 (0.2–0.7), 0.0023	Q1, 17–41	not found
*PTGER1*, *PTGER3*, *PTGER4*	UCES	2.3 (1.5–3.5), 0.00012	Median, NA–NA	not found
	UCES, grade 3	2.0 (1.2–3.3), 0.0075	Q1, 60–29	not found
*PTGDS*, *PTGDR2*	LUAD	0.4 (0.3–0.6), 4 × 10^−5^	Q1, 21–42	CESK, SARC
*TBXAS1*, *PTGDS*, *AKR1C3*, *PTGIS*, *CBR3*, *TBXA2R*, *PTGDR*, *PTGFR*, *PTGER3*, *PTGER4*	LIHC	2.2 (1.5–3.1), 1.5 × 10^−5^	Q1, 28–10	KIRP
*PTGES2*, *PTGES3*, *PRXL2B*, *AKR1C3*, *PTGES*	LIHC	2.2 (1.5–3.1), 1.3 × 10^−5^	Q1, 27–11	LUAD, PAAD
	LIHC, female gender	2.7 (1.5–5.0), 0.00059	Q1, 30–9	not found
	LIHC, male gender	2.4 (1.5–3.7), 9.2 × 10^−5^	Q1, 28–10	not found
	LIHC, grade 3	2.9 (1.5–5.8), 0.0014	Q1, 50–12	not found
	LIHC, high mutation burden	2.7 (1.6–4.4), 5.8 × 10^−5^	Q1, 60–14	not found
	LIHC, low mutation burden	2.0 (1.1–3.4), 0.015	Q1, 26–20	not found
*PTGIS*, *PTGDS*, *PTGES3*, *AKR1C3*, *CBR3*, *TBXA2R*, *PTGDR*, *PTGER1*, *PTGFR*, *PTGER3*	KIRP	5.1 (2.7–9.7), 3.4 × 10^−10^	Median, NA–NA	not found
	KIRP, low mutation burden	10.1 (3.6–28.3), 5.4 × 10^−8^	Median, NA–NA	not found

## Data Availability

Not applicable.

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
