# Peer review of "Prostanoid Signaling in Cancers: Expression and Regulation Patterns of Enzymes and Receptors"

_biology, 2022, doi:10.3390/biology11040590_

Round 1

Reviewer 1 Report

The authors identified the highly heterogeneous gene and protein expression landscape of prostanoid enzymes and receptors in different tumors and suggested the models of tumor-specific expression regulation. The consequence of the study is an assumption of prostanoid-dependent or prostanoid-independent tumor phenotypes, so targeting the prostanoid signalling could be an additional anticancer strategy. Albeit, the current study paves the way for more accurate therapeutics for cancer research. I still have some minor suggestions.

1, All figures are highly professional, and the authors should guide the readers to the meaning of the images appropriately; otherwise, it is likely to cause misunderstandings. Therefore, I suggest that the author consider revising these figure legends again.

2,  The authors analyzed the promoter methylation status of the target genes show the decrease of PTGDR, PTGER3, PTGIR, TBXA2R gene expression. It would be much better if the author can validate their data via other databases (https://biit.cs.ut.ee/methsurv/) in cancer (PMID: 29264942, 34834441, 33469338). 

3, The author demonstrated an association between overall patient survival with transcriptomics signatures in different types of tumors, including PTGIS, PTGES, TBXAS1. Since Connectivity Map (CMap) can be used to discover the mechanism of action of small molecules, functionally annotate genetic variants of disease genes, and inform clinical trials. It would be fascinating if these data could be correlated with other clinical databases. Therefore, I suggest the authors can validate their data via CMap or L1000 platform (PMID: 29195078, 32064155, 31888299).

4, Please try to avoid using unimaginably long sentences, such as Abstract (The results of the systematic….), some of the sentences in the manuscript even more than 30 words.

5, The font is too small for some of the current figures, and the figure legends also need proofreading.

Author Response

Reviewer 1

Comments and Suggestions for Authors

The authors identified the highly heterogeneous gene and protein expression landscape of prostanoid enzymes and receptors in different tumors and suggested the models of tumor-specific expression regulation. The consequence of the study is an assumption of prostanoid-dependent or prostanoid-independent tumor phenotypes, so targeting the prostanoid signalling could be an additional anticancer strategy. Albeit, the current study paves the way for more accurate therapeutics for cancer research. I still have some minor suggestions.

Authors: Thank you for the interest to our manuscript and valuable comments

Reviewer commentary:

All figures are highly professional, and the authors should guide the readers to the meaning of the images appropriately; otherwise, it is likely to cause misunderstandings. Therefore, I suggest that the author consider revising these figure legends again.

Authors response:

Corrected accordingly.

Figure 1 legend was changed from:

A scheme of prostanoid signaling: prostanoid-metabolizing enzymes (TBXAS1, PTGIS, PTGDS, PTGES, PTGES2, PTGES3, PRXL2B, AKR1C3, CBR1, CBR3); G-protein-coupled prostanoid receptors (TBXA2R, PTGIR, PTGDR, PTGFR, PTGDR2, PTGER1, PTGER2, PTGER3, PTGER4); prostanoids (TXA2 (thromboxane A2), TXB2 (thromboxane B2), PGE2 (prostaglandin E2), PGF2α (prostaglandin F2α), PGD2 (prostaglandin D2), PGI2 (prostacyclin), PGH2 (prostaglandin H2)). Subcellular localization: endoplasmic reticulum (TBXAS1, PTGIS, PTGDS, PTGES); cytosol (PTGES2, PTGES3, PRXL2B, AKR1C3, CBR1, CBR3); plasma membrane (all receptors). Membrane and endoplasmic reticulum image templates were obtained from https://smart.servier.com/.

to:

A scheme of prostanoid signaling: prostanoid-metabolizing enzymes (ellipses), G-protein-coupled prostanoid receptors (trapezes), type of G-subunit (circles) and prostanoids (rectangles). Proteins are shown according to their subcellular localization (endoplasmic reticulum, cytosol and plasma membrane). Proteins’ and prostanoids’ names correspond to the list of abbreviations. Membrane and endoplasmic reticulum image templates were obtained from https://smart.servier.com/.

We do not show the full names of proteins and prostanoids due to the fact that in this case the legend becomes overloaded with text and difficult to perceive.

Figure 2 legend was changed from:

A flowchart of data mining process using web-based bioinformatic tools. DEGs - dif-ferentially expressed genes; DEPs - differentially expressed proteins; DMGs - differentially methylated genes; CNVs - copy number variations; mutations - cancer-specific mutation frequency of target genes.

to:

A flowchart of data mining using web-based bioinformatic tools: GEPIA2, UALCAN and cBioPortal were used for analysis of The Cancer Genome Atlas (TCGA) and Clinical Proteomic Tumor Analysis Consortium data (CPTAC); WebGestalt – WEB-based GEne SeT AnaLysis Toolkit; hTFtarget – database for regulations of human transcription factors and their targets; CSmirTar – Condition-Specific miRNA Targets database; ONCOmir – OncoMir Cancer Database. ROC-plotter – ROC-plotter server; KM-plotter – Kaplan-Meier plotter server. Abbreviations: DEGs - differentially expressed genes; DEPs - differentially expressed proteins; DMGs - differentially methylated genes; CNVs - copy number variations; mutations - cancer-specific mutation frequency of target genes.

Figure 3 legend was changed from:

A landscape of differentially expressed genes, encoding prostanoid-metabolizing enzymes and receptors, in different tumors. Statistically significant changes (fold change cutoff = 2) in tumor/normal tissues are shown by arrows.

to:

A landscape of differentially expressed genes, encoding prostanoid-metabolizing enzymes (TBXAS1, PTGIS ,PTGDS ,PTGES, PTGES2, PTGES3, PRXL2B, AKR1C3, CBR1, CBR3) and receptors (TBXA2R, PTGIR, PTGDR, PTGFR, PTGDR2, PTGER1, PTGER2, PTGER3, PTGER4), in different tumors. Statistically significant changes (fold change cutoff = 2) in tumor/normal tissues are shown by arrows. “Groups” correspond to groups of tumors distinguished according to a similar pattern of DEGs. Up- and down-regulated genes are highlighted with orange and green colors, respectively. Genes’ and tumors’ names correspond to the list of abbreviations.

Figure 4 was divided in two parts: Figure 4 (part B) and Appendix B (part A). The Figure 4 legend was changed from:

Cluster analysis of differentially expressed genes, encoding prostanoid metabolizing enzymes and prostanoid receptors in tumors: (A) - Classification and Regression Trees method; (B) - Principal Component Analysis; color scale shows cluster distances.

to:

Principal component analysis of differentially expressed genes, encoding prostanoid metabolizing enzymes and prostanoid receptors, in different tumors; color scale shows cluster distances.

Figure 5 legend was changed from:

Comparative analysis of expression patterns of oncomiRs and genes, encoding prostanoid-metabolizing enzymes and prostanoid receptors, in different tumors.

to:

Comparative analysis of expression patterns of oncomiRs and genes, encoding prostanoid-metabolizing enzymes and prostanoid receptors, in different tumors. Genes’ names correspond to the list of abbreviations.

Reviewer commentary:

The authors analyzed the promoter methylation status of the target genes show the decrease of PTGDR, PTGER3, PTGIR, TBXA2R gene expression. It would be much better if the author can validate their data via other databases (https://biit.cs.ut.ee/methsurv/) in cancer (PMID: 29264942, 34834441, 33469338).

Authors response:

Agree.

The MethSurv (https://biit.cs.ut.ee/methsurv/) uses TCGA datasets as a data source like UALCAN does. These tools are different from each other. UALCAN focuses on the comparative analysis of normal and tumor tissue samples. MethSurv allows it to analyze the patients’ survival rates according to the methylation pattern of different regions of the specific gene locus. We also did not find in the MethSurv interface the option of comparison with normal tissue samples. We revealed that the primary data used by UALCAN and MethSurv do not always have the same identifier set of “CpG id” (MethSurv) / “Ilumina id” (UALCAN). UALCAN calculates an integral median of the values over the entire set of identifiers with which it works. MethSurv calculates “beta value” individually for each identifier. Therefore, it is not possible to directly compare the “beta values” calculated by these two tools. As for  PTGDR, PTGER3, PTGIR and TBXA2R genes, we obtained  beta values for the subsets of “CpG id” (MethSurv) corresponding sets of “Ilumina id” identifiers (UALCAN). The median “beta values” were calculated for each “CpG id” identifier. After that, the median and mean values within the subsets were estimated. It was shown that MethSurv’s median or mean values of “beta values” were comparable to the UALCAN integral medians of “beta values” for the corresponding subsets of identifiers (Table 1).

Table 1. Сomparison of “beta values”calculated by UALCAN and MethSurv web-based tools.

Gene-Tumor

“beta values”

UALCAN, median

MethSurv, mean value

MethSurv, median

PTGDR-COAD

0.218

0.20

0.16

PTGDR-UCES

0.494

0.48

0.49

PTGER3-COAD

0.261

0.28

0.20

PTGIR-KIRP

0,860

no CpG id / Ilumina id intersection

PTGIR-UCES

0.865

0.86

0.87

TBXA2R-LUAD

0.248

no CpG id / Ilumina id intersection

TBXA2R-LUSC

0.213

0.20

0.15

Reviewer commentary:

The author demonstrated an association between overall patient survival with transcriptomics signatures in different types of tumors, including PTGIS, PTGES, TBXAS1. Since Connectivity Map (CMap) can be used to discover the mechanism of action of small molecules, functionally annotate genetic variants of disease genes, and inform clinical trials. It would be fascinating if these data could be correlated with other clinical databases. Therefore, I suggest the authors can validate their data via CMap or L1000 platform (PMID: 29195078, 32064155, 31888299).

Authors response:

The Result’s section “Prognostic and predictive values of transcriptomic signatures” were divided into two sub-sections: “Prognostic value of transcriptomic signatures” and “Predictive value of prostanoid enzymes and receptors genes”. The Abstract was corrected accordingly.

The CMap database is focused more on the accumulation of transcriptomics profiling data of cell lines perturbed with several classes of chemical agents (drugs, metabolites, and biologically active compounds) depending on the dose and exposure time. CMap is a drug-oriented database and allows it to predict potential molecular targets and chemical agents based on gene expression patterns. “Perturbogens” do not fully cover the list of anticancer drugs, for example, fluorouracil and epirubicin are not represented in CMap. There are records for cyclophosphamide, paclitaxel, and docetaxel, but the expression patterns of genes in prostanoid signaling were not perturbed by these drugs. We additionally found potential drugs that might be associated with differentially expressed prostanoid enzymes (up and down-regulated genes) in COAD, LUSC, OV, READ, TGCT tumors from group III (Figure 3) using L1000CDS2 server (https://maayanlab.cloud/L1000CDS2/#/index). As we input a very limited number of genes to form a query (less than 10 genes), the value of output predictions (for example, antifungal drugs), albeit with overlap value = 0.5, is very modest. Alternative web-based tools such as Xena Browser (https://xenabrowser.net/) and PrognoScan (http://dna00.bio.kyutech.ac.jp/PrognoScan/index.html) are adapted for meta-analysis of different transcriptomic datasets, including CMap data. However, these tools are configured to search for associations with survival rates in respect to a single gene only from a multigene panel. Thus, we could not validate the tissue-specific transcriptomics signatures using the Cmap database, and the choice of the Kmplotter tool turned out to be most relevant in terms of coverage of more than one database.

Reviewer commentary:

Please try to avoid using unimaginably long sentences, such as Abstract (The results of the systematic….), some of the sentences in the manuscript even more than 30 words.

Authors response:

The simple summary and abstract as well as manuscript were corrected accordingly, thank you for pointing it out.

Reviewer commentary:

The font is too small for some of the current figures, and the figure legends also need proofreading.

Authors response:

Agree. We have modified Figure 4 by dividing it into two parts. The new version of Figure 4 presents data on clustering of differentially expressed genes (PCA method), while the presentation of results by CRT method is placed into Appendix B.

We have also modified Figure 3 by reducing the width of the cells, increasing the font of the labels, and deleting two columns for gene expression changes in two tumor types of group IV cancers. A group IV is “minor” because only 2 of 19 genes were found to be differentially expressed genes. The description of this group remains in the text.

Reviewer 2 Report

In the manuscript, the Authors performed a comparative systematic analysis of expression patterns of target genes, encoding prostanoid enzymes and receptors, and possible ways of their regulation at different levels in normal and tumor tissues. Assuming the presence of a highly heterogeneous gene and protein expression landscape of prostanoid enzymes and receptors in 24 different tumors and suggesting models based on tumor-specific expression regulation for prostanoid genes.

- Authors should rewrite the introduction, which seems more suitable for a review than an original article

- The Figure 1 and Table 1 are not needed, it is better to explain these literature data in the introduction, as support to the aim of the manuscript

- The manuscript reports only results obtained by bioinformatic analysis, which need an experimental validation in tumor tissues and matched normal tissues

Author Response

Reviewer 2

Comments and Suggestions for Authors

In the manuscript, the Authors performed a comparative systematic analysis of expression patterns of target genes, encoding prostanoid enzymes and receptors, and possible ways of their regulation at different levels in normal and tumor tissues. Assuming the presence of a highly heterogeneous gene and protein expression landscape of prostanoid enzymes and receptors in 24 different tumors and suggesting models based on tumor-specific expression regulation for prostanoid genes.

Authors: Thank you for the interest to our manuscript and valuable comments

Reviewer commentary

Authors should rewrite the introduction, which seems more suitable for a review than an original article

Authors response:

Agree. The text in the Introduction section has been corrected accordingly (please see the Text file with corrections). Figure 2 (a flow chart of data mining) and a piece of text describing it have been moved to the end of the Methods section.

Reviewer commentary

The Figure 1 and Table 1 are not needed, it is better to explain these literature data in the introduction, as support to the aim of the manuscript

Authors response:

We believe that Figure 1 should be left in the Introduction section, since it gives a visual representation of the “form and content” of prostanoid signaling and subcellular localisation of its participants for a wide audience of readers.

Table 1 was removed from the Introduction section and placed in Appendix A.

Reviewer commentary

The manuscript reports only results obtained by bioinformatic analysis, which need an experimental validation in tumor tissues and matched normal tissues

Authors response:

For the first time, we performed a large-scale bioinformatics analysis of experimental datasets which are available in public repositories (thousands of samples) on gene expression of prostanoid enzymes and receptors in solid tumors and normal tissues. To explain the observed expression patterns, we proposed models of regulation of gene expression at different levels. Also, links were found between disease prognosis and gene expression “panels” in different tumors. An experimental validation in the studies like this, even in a limited scope, would really increase the value of bioinformatic results. However, its absence does not reduce the scientific significance of data obtained in systems biology analysis and it is quite consistent with the objective of the study.

We added a text fragment (below) to the end of the Discussion section concerning the limitation of the study:

The limitations of the study are related to the use of publicly available data from TCGA, CPTAC and other repositories and web-based tools for analysis of datasets. The results of the study have more fundamental rather than translational value. The identified transcriptional signatures, with the participation of prostanoid signaling genes with differential expression in tumor/normal tissues, are exploratory. Thus, to further establish clinical relevance of such signatures, additional rounds of experimental validation should be required.

Authors:

Corrections were done in the Simple Summary, Abstract, Methods, Results and Conclusion sections of the manuscript. The changes are highlighted in the text.

Reviewer 3 Report

General comments to the paper entitled: Prostanoid signaling in cancers: expression and regulation patterns of enzymes and receptors

This is an excellent paper giving a comprehensive analysis of prostanoid signaling. It is well known that arachidonic acid pathways strongly influence cell growth. COX-2 gene inhibition is considered as having role in cancer prevention. The paper evaluates all the available data to find correlation on all level, gene expression, mutation, methylation, protein expression, post-translational modification, protein-protein interaction. The paper shows the complexity of the prostanoid signaling system, finding correlation between gene expression and prognosis but it is also concluded that hard to find “general” conclusion would be valid for all the 24 solid tumors.

I suggest a couple of minor changes

Use signaling instead of signalling.

In Table 1 I suggest to lift up to the first line the symbol of the Prostanoids.

The Figure 3 suggest four groups of tumors distinguished according to a similar pattern of differential expression of genes (DEF). I suggest rearranging the Fig 3. by clustering the genes belonging to the same group (I-II-III-IV).

Author Response

Reviewer 3

Comments and Suggestions for Authors

General comments to the paper entitled: Prostanoid signaling in cancers: expression and regulation patterns of enzymes and receptors

This is an excellent paper giving a comprehensive analysis of prostanoid signaling. It is well known that arachidonic acid pathways strongly influence cell growth. COX-2 gene inhibition is considered as having role in cancer prevention. The paper evaluates all the available data to find correlation on all level, gene expression, mutation, methylation, protein expression, post-translational modification, protein-protein interaction. The paper shows the complexity of the prostanoid signaling system, finding correlation between gene expression and prognosis but it is also concluded that hard to find “general” conclusion would be valid for all the 24 solid tumors.

Authors: Thank you for the interest to our manuscript and valuable comments

I suggest a couple of minor changes

Reviewer commentary:

Use signaling instead of signalling.

Authors response:

Corrected accordingly.

Reviewer commentary:

In Table 1 I suggest to lift up to the first line the symbol of the Prostanoids.

Authors response:

Аgree. We had to accept another reviewer's suggestion and moved Table 1 from the Introduction to Appendix A.

Reviewer commentary

The Figure 3 suggest four groups of tumors distinguished according to a similar pattern of differential expression of genes (DEF). I suggest rearranging the Fig 3. by clustering the genes belonging to the same group (I-II-III-IV).

Authors response:

We have modified Figure 3. Тwo columns corresponding to tumor types (HNSC, LGG) of group IV were deleted. This group is “minor” because only 2 of 19 genes were found to be differentially expressed genes. The description of group IV remains in the text.

Round 2

Reviewer 2 Report

None